# Enrichment of 3D-Printed k-Carrageenan Food Gel with Callus Tissue of Narrow-Leaved Lupin *Lupinus angustifolius*

**DOI:** 10.3390/gels9010045

**Published:** 2023-01-06

**Authors:** Kseniya Belova, Elena Dushina, Sergey Popov, Andrey Zlobin, Ekaterina Martinson, Fedor Vityazev, Sergey Litvinets

**Affiliations:** 1Institute of biology and biotechnology, Vyatka State University, 36, Moskovskaya Str., 610000 Kirov, Russia; 2Institute of Physiology of Federal Research Centre “Komi Science Centre of the Urals Branch of the Russian Academy of Sciences”, 50, Pervomaiskaya Str., 167982 Syktyvkar, Russia

**Keywords:** lupin, callus culture, k-carrageenan, 3D food printing, gel, texture, rheology, simulated digestion, phenolic compounds, scanning electron microscopy

## Abstract

The aim of the study is to develop and evaluate the printability of k-carrageenan inks enriched with callus tissue of lupin (*L. angustifolius*) and to determine the effect of two lupin calluses (LA14 and LA16) on the texture and digestibility of 3D-printed gel. The results demonstrated that the enriched ink was successfully 3D printed at concentrations of 33 and 50 g/100 mL of LA14 callus and 33 g/100 mL of LA16 callus. The feasibility of 3D printing is extremely reduced at higher concentrations of callus material in the ink. The hardness, cohesiveness, and gumminess of the 3D-printed gel with LA16 callus were weakened compared to the gel with LA14 callus. The results of rheological measurements showed that an increase in the content of LA16 callus interfered with the formation of a k-carrageenan gel network, while LA14 callus strengthened the k-carrageenan gel with increasing concentration. Gel samples at different concentrations of LA14 and LA16 calluses formed a spongy network structure, but the number of pores decreased, and their size increased, when the volume fraction occupied by LA14 and LA16 calluses increased. Simple polysaccharides, galacturonic acid residues, and phenolic compounds (PCs) were released from A-FP gels after sequential in vivo oral and in vitro gastrointestinal digestion. PCs were released predominantly in the simulated intestinal and colonic fluids. Thus, incorporating lupin callus into the hydrocolloid ink for food 3D printing can be a promising approach to developing a gelling material with new mechanical, rheological, and functional properties.

## 1. Introduction

Plant-based foods are essential for a healthy diet, and they are growing in popularity as their positive effects on human health gain wider recognition [1]. However, it is becoming more difficult to provide plant food to the constantly growing population of the planet [2]. New technologies need to be developed to produce varied and healthy plant-based foods. The cultivation of plant cells as a new approach to the production of plant-based food can become such a technology. A number of food ingredients have been registered and are currently being produced using plant cell cultures [3,4]. The possibilities of obtaining a number of valuable nutrients using cell cultures have been studied in some detail, and approaches have been developed for enriching food with ingredients beneficial to health.

Three-dimensional (3D) printing opens new possibilities for creating complex geometric structures in the individual production of dishes based on the specific nutritional needs and calorie intake of the individual [5]. The new opportunity provided by food 3D printing is extremely relevant for people with special nutritional needs, such as the elderly or patients who have difficulty eating or swallowing, children, and people with various metabolic and inflammatory disorders. This technology is easy to operate, allows for mass production, and also provides economic and environmental benefits by reducing food waste and financial costs for storing and transporting food [6]. Basically, for 3D printing food products, the extrusion method is used, in which a liquid or semi-solid material is extruded through a nozzle, creating a three-dimensional structure layer by layer, according to the digital model. Mashed potatoes, chocolate, cheese, meat, surimi, vegetable, and fruit pastes can be used as food printing materials [7]. However, with extrusion printing, it is not always possible to obtain a programmed 3D design. The decisive factor determining the quality of the printed product is the physicochemical properties of the food-grade inks (rheological and thermal properties, water retention capacity, and specific mechanisms of aggregation and gelation) [8]. Thus, both plant cell cultures and 3D printing are increasingly being used in food production. However, there is little work on the inclusion of plant cell cultures in 3D printing, although the number of original studies on plant-based 3D-printable materials has increased significantly in the past few years [9]. The question of how the inclusion of plant cell cultures can change the rheological properties of food ink and determine the structural and mechanical characteristics of the resulting 3D product remains poorly understood.

Carrot callus cells have been used as an ingredient for food 3D printing [10]. In this study, callus-based edible ink was used based on 4% alginate, the gelation of which requires divalent cations [11]. Therefore, callus-alginate ink requires curing using Ca^2+^ ions to form a hard gel after 3D printing. In the present study, we hypothesized that carrageenan-based inks would be suitable for 3D food inks, as carrageenan is a food-grade hydrocolloid with good gelling properties [12]. The callus tissue of *Lupinus angustifolius*, a narrow-leaved lupin, was chosen to demonstrate the applicability of callus material in 3D printing. *Lupinus* species is a genus of a widely consumed leguminous plant of the Fabaceae family. Several health-promoting properties of Lupinus species, including *L. angustifolius*, have been reported in preclinical and clinical human and animal studies, including antioxidant, anti-inflammatory, hypolipidemic, hypoglycemic, and hypotensive properties, among others. These biological activities are attributed to their human-health-beneficial chemical components, such as polyphenols, carotenoids, and other phytochemicals [13]. These properties make the development of cell cultures of lupin for functional food products promising.

The study’s purpose was to develop the proposed k-carrageenan ink for 3D printing with callus tissue from narrow-leaved lupin, *L. angustifolius*, evaluate its printability, and determine the effect of callus on the texture and digestibility of 3D-printed gel.

## 2. Results and Discussion

### 2.1. Characterization of Lupin Callus

The cell biomass of two lines of lupine callus was used to obtain food ink for 3D printing. Callus LA14 was grown in a medium containing 2,4-dichlorophenoxyacetic acid (2,4-D) and 6-benzylaminopurine, whereas callus LA16 was grown in a medium containing kinetin and naphthylacetic acid. Callus LA14 had loose, moderately wet grayish-white tissue (Figure 1A), while the LA16 callus tissue was less loose and consisted of cells of a browner color than the LA14 tissue (Figure 1B).

Callus LA14 consisted of rounded, thin-walled parenchymal cells with an average size of 129 ± 48 µm. LA16 callus cells had an elongated and curved shape with an average size of 166 ± 116 μm (Figure 2A,B). The sphericity factors of callus cells LA14 and LA16 were 0.24 ± 0.11 and 0.61 ± 0.10 (*p* < 0.05), respectively. 

The LA16 callus had 1.6 times the soluble solid content (SSC) of the LA14 callus (Table 1). LA14 and LA16 were similar in protein content but had different polysaccharide compositions. Hardly hydrolyzable polysaccharides predominated in both calluses, which included mainly cellulose and some hemicelluloses. The content of easily hydrolyzable polysaccharides (EHP), which included pectins, arabinogalactans, etc., was 1.6 times higher in LA16 than in LA14. The content of phenolic compounds (PCs) was 2.4 times lower in LA16 than in LA14 (Table 1). The residual content of 2,4-D in LA14 callus was 0.88 mg/kg, which is unacceptable under food safety requirements. Therefore, LA16 callus was considered a promising food ink ingredient for the rest of the study, while LA14 callus was used only for comparison.

### 2.2. 3D Printing

Three ink formulations, labeled LA14-33, LA14-50, and LA14-66, were prepared by mixing 33, 50, and 66 g of LA14 callus biomass with 3 g of k-carrageenan dispersed in 100 mL of peach juice. Food inks prepared with 33, 50, and 66 g/100 mL of EA16 callus were labeled as LA16-33, LA16-50, and LA16-66, respectively. The formulations were heated to 90 °C and loaded into a food capsule for a 3D printer. After extrusion from the capsule, the ink solidified on the platform in the form of a 3D gel structure. The printability was qualitatively evaluated in terms of ease and uniformity of extrusion, precision and accuracy of printing, and stability of the geometry after printing. The use of LA14-33 and LA14-50 inks allowed for both easy extrusion from the nozzle and good shape retention even under the load of 30 successive layers (Figure 3A,B). However, the LA14-66 ink could not be 3D printed. First, the thicker ink of LA14-66 clogged the capsule nozzle, causing the ink to be squeezed out unevenly during printing; towards the end of printing, the capsule nozzle became completely clogged, and printing stopped. Second, the structure printed with LA14-66-containing ink collapsed once printed (Figure 3C).

Representative results of the printing experiment with food inks containing LA16 callus are shown in Figure 4. The consistency of LA16-33 ink allowed 3D printing to start, but printing stopped due to an ink clot in the capsule nozzle, so the 3D object was not printed completely (Figure 4A). 3D printing with LA16-50 and LA16-66 inks was disturbed due to the formation of thick portions that were squeezed out unevenly from the capsule nozzle. It was not possible to finish printing with LA16-50 and LA16-66 inks (Figure 4B).

### 2.3. Rheological and Mechanical Properties of 3D-Printed Gels

Angular frequency sweep measurements are conducted to predict the structural integrity and mechanical strength of a 3D-printed material. All of the 3D-printed samples, excluding LA16-66, exhibited a greater storage modulus (G′) value than loss modulus (G″) (Figure 5). This suggests the network integrity of molecules in carrageenan-based gel samples. The obtained high G′ values (40,000 > G′ > 90,000 Pa) and middle elastic character (tan δ ≤ 1.0) demonstrated that all 3D-printed gels, with the exception of the LA16-66 sample, were strong physical gels (Figure 5D).

The rheological properties of the 3D-printed gel depended on the type of callus it contained. The values of G′ and G″ and complex viscosity increased with an increase in the content of LA14 in the ink (Figure 5A and Table 2). However, the values of G′ and G″, as well as the values of complex viscosity, decreased with an increase in the content of LA16 callus (Figure 5B and Table 2). For the LA16-66 sample, G′ was lower than G″ at an oscillation frequency of less than 15 Hz, indicating that the sample was a transitional sol-gel phase rather than a gel. G′ became greater than G″ at oscillation frequencies greater than 15 Hz, indicating that the LA16-66 sample behaved as a non-Newtonian fluid whose viscosity is known to change to become more solid at high oscillation frequencies. The non-Newtonian behavior of the LA16-66 sample appeared to be due to the large number of callus cells that it contained.

The mechanical behavior of the 3D-printed gels in a double compression test at room temperature is presented in Figure 6. The parameters evaluated in this study were hardness, springiness, cohesiveness, and gumminess. The hardness is the peak force during the first compression cycle and is related to the strength of the gel’s structure when it is under compression. The hardness of the LA-14 callus-containing gels increased with the increase in callus tissue content (Figure 6A). The hardness of the LA14-66 gel was 1.5 times higher than that of the LA14-30 and LA14-50 gels. After the deformation of the sample, the springiness (extent to which the sample springs back after it has been deformed during the first compression) was measured. The springiness of the LA14 gel decreased slightly as the amount of callus in the ink increased (Figure 6B). The next parameter to be measured was cohesiveness, which is related to how well the gel withstands a second deformation relative to its resistance under the first deformation. LA14-50 gel exhibited the highest levels of cohesiveness. The cohesiveness of the LA14-33 and LA14-66 gels was 5 and 46% lower than that of the LA14-50 gel (Figure 6C). Lastly, gumminess, which indicates the amount of mastication needed to make a food item ready to swallow, was the same regardless of the content of LA14 callus in the gel (Figure 6D).

The values of the textural parameters of the LA16 gels were significantly lower than those of the LA14 gels. Increasing the callus content resulted in a decrease in the hardness of the LA16 gel such that the hardness of the LA16-66 gel was two times lower than the hardness of the LA16-50 gel and 5.4 times lower than the hardness of the LA14-66 gel (Figure 6A). The springiness of the LA16 gel decreased slightly when the callus content was increased from 50 to 66 g/100 mL (Figure 6B). The cohesiveness of the LA16 gel decreased 2.9 times with an increase in callus content from 50 to 66 g/100 mL (Figure 6C). The gumminess of the LA16-33, LA16-50, and LA16-66 gels was the same and was lower by 6, 4, and 3 times than the gumminess of LA14-33, LA14-50, and LA14-66 gels, respectively (Figure 6D).

### 2.4. Microstructure of 3D-Printed Gels 

Figure 7 shows the morphology of 3D-printed gels. Gel samples at different concentrations of LA14 and LA16 callus formed a spongy network structure, but the callus content notably affected sample morphology. The number of pores decreased and their size increased when the volume fraction occupied by calluses LA14 and LA16 increased. Callus inclusion in the gel seemed to hinder the ice crystals’ growth during the freezing process since the porosity formed when the water content was pulled away during the freeze-drying process. 3D-printed gels containing LA16 callus had smaller pores than gels containing LA14 callus. In the structure of the gel containing LA16-66 callus, dense, folded formations were found (Figure 7F). These formations can be clearly seen in the micrographs obtained using magnifications of 500× and 1000× (Figure 8). It can be assumed that these are dense conglomerates of callus cells.

EDS analysis revealed similar contents of oxygen (87–92 wt%) in the 3D-printed gels containing LA14 and LA16 calluses. It has been established that the content of the K^+^ cation in the gel increased with an increase in the concentration of callus in the ink. The K^+^ cation content was 3.9 ± 0.0, 5.0 ± 0.2, and 6.5 ± 0.0 wt% in LA14-33, LA14-50, and LA14-66 gel samples, respectively. The K^+^ cation content was 4.7 ± 0.5, 5.1 ± 0.4, and 7.5 ± 0.3 wt% in LA16-33, LA16-50, and LA16-66 gel samples, respectively.

### 2.5. Simulated Digestibility of 3D-Printed Gel Containing LA16-33 Callus

The content of SSC in the incubation medium after each phase of simulated digestion gradually decreased, indicating good digestibility of the 3D-printed gel (Figure 9A). The incubation medium after each phase of simulated digestion of LA16-33 gel contained PCs. The most PCs were released in the SIF and SCF media. Two times fewer PCs were released during the destruction of the gel during in vivo chewing and during the subsequent in vitro SGF phase than during the SIF and SCF phases (Figure 9B). The content of PCs in the undigested residue was 21 ± 7 μg/mL.

During in vivo chewing (OP), the gel containing LA16-33 released the most sugars. The content of neutral sugars in the incubation medium was 146 ± 20, 81 ± 7, 45 ± 4, and 22 ± 5 mg/mL after the OP, SGF, SIF, and SCF phases, respectively. The main neutral sugars that were released during digestion were fructose and glucose (Figure 10). Sucrose was not detected in phases OP, SGF, and SIF, but it was contained in the SCF incubation medium at a concentration of 6.6 ± 1.0 mg/mL. In addition to neutral monosaccharides, residues of galacturonic acid were found in the digestion medium.

The present study was devoted to the enrichment of hydrocolloid ink with callus material for 3D-printed food gel. Inks for 3D printing were prepared from a mixture of 3% k-carrageenan and biomass of lupine calluses LA14 and LA16. The results demonstrated that the enriched ink was successfully 3D printed at concentrations of 33 and 50 g/100 mL of LA14 callus and 33 g/100 mL of LA16 callus.

k-carrageenan is a linear sulfated polysaccharide derived from red algae, which is widely used in the food industry due to its ability to form thermoreversible gels [12]. In 3D food printing, k-carrageenan has been used as part of multicomponent food inks along with gelatin [14], starch and xanthan [15], beeswax and xanthan [16], freeze-dried vegetable powders [17], surimi [18], and others. However, unless the 3D printing parameters were changed, one-component k-carrageenan ink was only marginally printable [19,20]. Based on existing data, a 3% solution of k-carrageenan did not allow us to obtain 3D objects (data not shown). The mechanism of k-carrageenan gelation involves a conformational transition from a random coil structure when heated to double helices, followed by the aggregation of helices during the cooling process [21]. The reason for the low printability of k-carrageenan is that the transition from random coils to double helices is extremely fast, whereas the subsequent aggregation of double helices to complete the formation of a gel network takes several hours [19]. The 3% k-carrageenan ink supplemented with lupin calluses was successfully 3D printed. Therefore, the addition of callus materials seemed to accelerate the gelation of k-carrageenan. This effect may be related to the gel-promoting action of ions and polysaccharides contained in the callus. It has been found that as the content of callus increased, printing deteriorated due to the thickening of the ink in the printing capsule and difficulty in extrusion. At the same time, there was an increase in the content of potassium ions in the ink. It is well known that the gelation of carrageenan is highly dependent on potassium ion concentration [22]. The printability of ink with LA16 callus at concentrations of 50 and 66 g/100 mL was lower than that of LA14 callus, possibly due to the higher content of polysaccharides in LA16 callus. Indeed, the content of easily hydrolyzable polysaccharides in LA16 callus was 1.6 times higher than in LA14 callus. One of the main easily hydrolyzable polysaccharides is pectin [23], which is a gel-forming polysaccharide of the plant cell wall [24]. The content of pectin in plant callus was previously shown in a number of papers [25,26,27,28]. Therefore, we hypothesize that callus polysaccharides may alter the printability of carrageenan inks. However, the composition and gelling properties of lupine callus polysaccharides were not established in the present study.

The effect on the textural, rheological, and morphological properties of a 3D-printed gel was found to depend on the type and concentration of callus in the ink composition. Rheological measurements and a double compression test showed that the structure of the 3D-printed gel changed with increasing LA14 callus content. The increase in gel strength resulting from the addition of LA14 callus may be related to the gel-promoting action of K^+^ ions and polysaccharides contained in the callus. Carrageenans are formed of alternate disaccharide units of D-Gal and 3,6-anhydro-Gal joined by α-1,3- and β-1,4-glycosidic linkage; hydrogen bonding is well known to be the primary mechanism involved in carrageenan gelation (Figure 11A). Kappa-carrageenan contains one sulfate group (OSO_3_^−^) per disaccharide repeating unit; therefore, mono- and divalent cations cross-link double helices with outward-oriented OSO_3_^−^ groups in addition to hydrogen bonding [29]. K^+^ ions from callus material are supposed to provide ionic cross-links between adjacent κ-carrageenan chains formed by sulfated galactose residues (Figure 11B, K^+^). Gelling polysaccharides of the callus cell wall, such as pectins, could also enhance the 3D-printed gel (Figure 11B, CPs). The size and shape of LA14 callus cells evidently did not interfere with the formation of cross-links (Figure 11B, LA14 callus cells). In the case of the LA16-containing ink, callus cell conglomerates may have prevented the convergence of the к-carrageenan chains and the formation of a three-dimensional network (Figure 11B, LA16 callus cells). As a result, a less strong gel was formed compared to the ink that contained LA14 callus. Further, the 3D object that was printed with the highest concentration of LA16 (66 g/100 mL) was not a “gel” and had the lowest hardness. We hypothesize that the elongated shape of LA16 callus cells may have contributed to the formation of cell conglomerates, the presence of which can be inferred from SEM images of gels printed with LA16 callus inks (Figure 6).

The effect of callus cells on the mechanical properties of the gel matrix was previously shown in the study [10], where the multiplication of carrot callus cells in agar gel reduced the gel strength of the fabricated structure. The authors considered that the decrease in the homogeneity of the gel matrix affected its mechanical properties. The encapsulation of living lettuce leaf cells in pectin-based bio-ink had a negative effect on mechanical properties [30]. A decrease in product hardness with the addition of cellular material has previously been shown when incorporating microalgae powder into cookies by conventional methods [31]. Conversely, Vieira et al. [32] incorporated different powder amounts of microalgae into shortbread biscuits, which promoted an increase in cookie hardness directly proportional to the microalgae powder content. A decrease in the hardness of the gel containing LA16 callus was accompanied by an increase in cohesiveness and gumminess. Similar mechanical behavior was observed when button mushroom powder was added to wheat dough [33].

The mechanical behavior of food gels during oral processing, such as chewing, bolus formation, and swallowing, is determined by their textural and rheological properties [34]. Furthermore, hardness, cohesiveness, gumminess, and springiness are critical parameters for predicting the sensory perception of food gels. Consumers consider hardness to be an important indicator because it can simulate the force required for food to compress between teeth or between the tongue and palate [35]. Softening the k-carrageenan gel induced by LA16 incorporation would presumably make it easier to chew. Therefore, LA16-containing gel may be required by patients with dysphagia [36]. Semi-solid foods with low hardness and high cohesiveness values are assumed to be ideal for dysphagic diets [37]. It has been shown that gel containing LA16 callus at a concentration of 66 g/100 mL had a low hardness and a high cohesiveness. However, the feasibility of this ink composition for 3D printing was low. Therefore, further modification of the lupin callus ink composition and printing parameters is required to improve fluidity during 3D printing.

In terms of nutritional quality, the lupin calluses contained a high concentration of polysaccharides and PCs. In order to preliminarily assess the nutritional value of the 3D gel containing LA16-33 callus, it was subjected to simulated digestion. Three-dimensional gels containing LA14 callus were excluded from this experiment due to food safety reasons—callus LA14 was grown in a medium containing 2,4-D, and the residual content of 2,4-D in LA14 callus was detected, which is not allowed in food products.

Successive oral in vivo and gastrointestinal in vitro digestion of LA16-33-containing gel in SGF, SIF, and SCF fluids demonstrated a decrease in the SSC content during digestion and the release of simple sugars. The data indicate good digestibility of the printed gel; however, it should be noted that some of the simple sugars come from peach juice, which was part of the ink. The detection of GalA in the incubation medium indicates partial hydrolysis of the polysaccharides of the callus cell wall. GalA is the main monosaccharide residue that forms the pectin macromolecule. Pectin is a complex plant polysaccharide that is resistant in the human stomach and small intestine and is fermented by colonic bacteria in the large bowel. So, pectin is a dietary fiber with good prebiotic properties [38,39]. Furthermore, pectin is highly valued as a functional food ingredient due to its hypolipidemic, hypoglycemic, satiating, antibacterial, and antitumor biological effects [40]. Therefore, it can be assumed that LA16-containing food gel will be a good source of pectin with health-promoting properties.

PCs represent secondary metabolites of plants and are highly valuable food components [41]. Many studies have shown that callus cultures are a good source of PCs [42,43,44,45,46]. PCs possess diverse bioactivities, including antioxidant, anti-inflammatory, and anticancer effects [47,48,49]. To predict the part of the gastrointestinal tract where LA16 callus would bring biological activity, PCs released from LA16-33 gels were evaluated during successive OP and SGF, SIF, and SCF digestion. The preferential release of PCs in the SIF and SCF indicates that digesting LA16-33 gel will preserve its biological potential until the lower intestines. Therefore, lupine callus may be of interest for the development of food gels for the prevention and treatment of inflammatory bowel diseases (IBD), which has emerged as a public health challenge worldwide. IBD is characterized by segmental inflammation anywhere in the intestine (Crohn’s disease) or superficial inflammation of the mucosal layer of the colon (ulcerative colitis) [50]. PCs have been shown to confer symptomatic and health-related quality of life improvements in IBD patients [51].

## 3. Conclusions

In this study, lupin calluses LA14 and LA16 were efficiently incorporated into k-carrageenan-based food ink for 3D printing. The printability of k-carrageenan ink with LA14 callus was higher than that with LA16 callus. 3D-printed food gel can contain up to 50 g per 100 mL of LA14 callus and 33 g per 100 mL of LA16 callus. The feasibility of 3D printing is extremely reduced at higher concentrations of callus material in the ink. The hardness, cohesiveness, and gumminess of 3D-printed gels with LA16 callus were weakened compared to gels with LA14 callus. The results of rheological measurements showed that an increase in the content of LA16 callus interfered with the formation of a k-carrageenan gel network, while LA14 callus strengthened the k-carrageenan gel with increasing concentration. The digestibility of 3D-printed gel with LA16 callus in a simulated digestion model showed its nutritional value as a source of simple sugars, cell wall polysaccharides, and PCs. Wherein PCs were released mainly into the SIF and SCF, demonstrating the biological potential for the lower intestines. Based on its combination of textural properties and digestive profile, lupine callus LA16 may be considered an ingredient in functional food gels for patients with dysphagia and intestinal inflammation. Incorporating lupin callus into the hydrocolloid ink for food 3D printing can be a promising approach to developing gelling material with new mechanical and rheological properties.

## 4. Materials and Methods

### 4.1. Materials

Refined KA120R k-carrageenan was purchased from Greenfresh Food Co., Ltd. (Longhai City, Zhangzhou, Fujian, China). The (-SO_3_-) content in k-carrageenan was 11.7 ± 0.7%. Peach juice concentrate was purchased from a local supermarket. The reagents, including the Folin and Ciocalteu’s phenol reagent, pectinase from *Aspergillus niger* (>1 U/mg), D-(+)-galacturonic acid monohydrate (99.1%), and ferulic acid (99.8%) were purchased from Sigma-Aldrich (St. Louis, MO, USA). To build calibration curves, we used: 2,4-dichlorophenoxyacetic acid (GSO 9105-2008, Ecolan, Russia), D-(+)-glucose monohydrate (99.9%; Panreac Química SLU, Barcelona, Spain); and 3,5-dimethylphenol (99.8%, Acros Organics, Waltham, MA, USA).

### 4.2. Obtaining and Characterizing Calluses

The callus culture of narrow-leaved lupine (*Lupinus angustifolius* L.) of the legume family (Fabaceae) was obtained in the laboratories of Vyatka State University’s Department of Biotechnology. Callus line No. 14 (LA14) was bred from stem segments of a native plant. Callus was induced on Murashige-Skoog medium with the addition of phytohormones: 2,4-dichlorophenoxyacetic acid (2,4-D)–2 mg/L and 6-benzylaminopurine (6-BAP)–0.2 mg/L. The optimal combination of phytohormones to support the growth of angustifolia callus during long-term cultivation is a combination of 2,4 D, 2.0 mg/L + 6-BAP, 0.1 mg/L. During several passages, lupine calli were grown on media with various combinations of phytohormones, with a visual assessment of cell growth and viability. The best transition from medium No. 14 to medium No. 16 (LA16) was observed with the following phytohormone contents: 1.0 mg/L naphthylacetic acid (NAA) and 0.1 mg/L kinetin.

The specific growth rates of calluses LA14 and LA16 were 0.46 ± 0.04 and 0.46 ± 0.04 g/d, respectively. The callus cultures were subcultured for 21 days at 26 °C in the darkness. A freshly harvested callus was microscopically examined and chemically analyzed. A portion of the harvested biomass was frozen for the subsequent preparation of ink for 3D printing.

The size and shape of the cells were studied using a Motic BA300 optical microscope (China) with a built-in video eyepiece. Each analysis was performed 10 times under the same optical conditions. Cell shape was determined quantitatively using dimensionless shape indicators. The sphericity factor (SF) was calculated using the following formula:SF = (dmax − dmin)/(dmax + dmin),(1)
where dmax is the maximum diameter (µm), and dmin is the minimum diameter perpendicular to dmax (µm).

The content of dry matter in the plant raw materials was determined by the thermogravimetric method. Protein content in callus tissues was determined according to Barnstein’s method.

The content of easily and hardly hydrolyzable polysaccharides was determined using the Kizel and Semiganovsky method. A sample was first prepared for the analysis of easily hydrolyzable polysaccharides: 100 mL of 2% HCl was added to 2.5 g of dry callus tissue, the resulting mixture was boiled on a stove in a flask under reflux for 3 h, the contents of the reaction flask were filtered through a paper filter, the filtrate and the washings were combined in a 250 mL volumetric flask, and the residue on the filter was used. After cooling, both samples were thoroughly mixed, and the concentration of reducing substances in them was determined by the ebulliostatic method of Nizovkin and Emelyanova by titrating copper-alkaline solutions with the obtained samples. The titer of copper-alkaline solutions was determined by glucose.

The concentration of reducing substances in the analyzed samples was calculated as a percentage of glucose according to the formula:X = T × n × 100/V × 100,(2)
where X is the concentration of sugar in the analyzed solution, %; T is the titer of the copper-alkaline solution for glucose, mg; V is the volume of the analyzed solution used for titration, mL. The content of EHP was calculated as follows:XL = CL × VL × k × 100/m × 100,(3)
where m is the mass of absolutely dry plant tissue taken for analysis, g; CL is the concentration of reduced substances in the hydrolyzate of EHP, %; VL is the volume of the hydrolyzate, mL; and k is the coefficient of conversion of monosaccharides into polysaccharides, which is 0.89.

The content of HHP was calculated by the formula:XT = CT × VT × n × K × 100/m × 100,(4)
where CT is the concentration of reducing substances in the diluted neutralized hydrolysate; VT is the volume of acid hydrolysate, in mL; n is the dilution of the hydrolysate during neutralization; K is the conversion factor for monosaccharides in HHP, which is 0.9.

For PC content in callus tissue, 200.0 ± 0.1 mg of freeze-dried callus tissue was placed in a 25-mL test tube, to which 5.0 mL of a 70% methanol solution was added, closed with a cork, and mixed on a vortex-type shaker. The test tube was placed in a water bath at a temperature of 70 °C for 10 min, stirring the contents on a shaker after 5 and 10 min. The tube was then cooled to room temperature and centrifuged for 10 min at 9000 rpm. The supernatant was decanted into a 25 mL graduated tube. The extraction of polyphenols with a 70% methanol solution was carried out twice more (in 5 mL portions). The extracts were combined, the extraction volume in a graduated test tube was adjusted to 15.0 mL with a 70% methanol solution, and it was kept in a refrigerator at 4 °C for 24 h [52]. A sample of 0.4 mL of the extract was transferred into a test tube, and 4.0 mL of a 0.2 M NaHCO3 solution in 0.1 M NaOH was added. Folin–Ciocalteu reagent, mixed for 10 min at room temperature. A sample of 0.4 mL of the Folin–Ciocalteu reagent was added to the solution, mixed, and incubated for 30 min in the dark at room temperature. The absorbance of solutions was measured in a cuvette with an optical path length of 10 mm at a wavelength of 765 nm. The concentration of polyphenols in the solution was determined from a calibration curve constructed using ferulic acid (10–100 μg/mL)

Quantitative determination of 2,4-D in callus extracts was determined in the form of methyl ester by chromato-mass spectrometry on a G2589A gas chromatograph (Agilent Tech., Santa Clara, CA, USA) with an HP-1MS capillary column (0.25 mm × 30 m) (Hewlett-Packard, Palo Alto, CA, USA). The carrier gas was helium (1 mL/min). The volume of the injected sample was 1 µL (flow split 10:1). The evaporator temperature was 250 °C. Temperature conditions of the column thermostat: 50 °C (0.5 min) → 100 °C (gradient–25 °C/min) → 220 °C (gradient–12 °C/min). Mass spectrometer 5973 INERT (Agilent Tech., Santa Clara, CA, USA): ionization chamber temperature, 230 °C; the energy of ionizing electrons was 70 mV. Quantitative determination of 2,4-D was carried out by extracted ions (*m*/*z*: 175.00; 199.00; 234.00) using the method of absolute calibration using standard solutions of 2,4-D methyl esters (0.025–0.50 mg/kg).

### 4.3. Inks Preparation and 3D Printing 

To obtain the ink ingredient, the frozen callus biomass was thawed and passed through a metal sieve with a mesh size of 1 mm to eliminate cell aggregates. Callus biomass (33, 50, or 66 g) and 3 g of k-carrageenan were dispersed in 100 mL of peach juice and left for one hour with magnetic stirring. The dispersion was heated to 90 °C in a slow cooker for 20–25 min. The hot solution was vigorously shaken and transferred into a printing capsule for a 3D printer. A FOODINI (Natural machines, Barcelona, Spain), an extrusion-based commercial 3D food printer was used to print 3D samples of different callus-containing inks. For the experiments in this study, a nozzle size of 1.5 mm was used. To assess the printability of the food inks, a “flower tower” of 30 layers was printed on a silicone mat. When the printed samples were able to maintain the structure for at least 15 min, they were considered printable. For rheological and textural measurements, samples in the form of a plate 1 cm × 1 cm × 0.3 cm (width, length, and height), and cube-shaped samples with a side of 5 mm were printed separately.

### 4.4. Measurement of Mechanical Properties

For two-cycle compression tests, gel samples (5 mm height, 5 mm length, and 5 mm width) were placed on the platform of the texture analyzer (Texture Technologies Corp., Stable Micro Systems, Godalming, UK). The tests were performed using a cylindrical aluminum probe P/25 (25-mm diameter). The gels were compressed twice at room temperature. The pre- and post-test speed was 5.0 mm/s and the test speed was 1 mm/s until a 100% strain. Destructive 100% strain was used to represent gel behavior during the chewing process. Eight replicates were made for each type of gel. For two cycles, compression-decompression provided a force-time graph and led to the extraction of eight parameters: hardness, cohesiveness, springiness, gumminess, chewiness, and resilience. All calculations were performed using Texture Exponent 6.1.4.0 software (Stable Micro Systems, Godalming, UK) according to the manufacturer’s instructions. One-way ANOVA with Tukey’s honest significance test was applied to determine statistically significant differences. Values of *p* ≤ 0.05 were considered statistically significant.

### 4.5. Measurement of Rheological Properties

The rheological property of the samples was determined in a rotational-type rheometer (Anton Paar, Physica MCR 302, Graz, Austria) equipped with a parallel plate geometry (diameter 25 mm) and a gap of 4 mm between the two plates. Four repetitions were performed for each sample. The sample loading area was preheated to 20 °C before gel loading. After loading, the samples were equilibrated at 20 °C for 5 min before the measurement. The obtained mechanical spectra were characterized by the values of G′ and G″ (Pa) as a function of frequency in the range of 0.05–50.00 Hz at 20 °C and a constant stress of 9.0 Pa, which was within the linear viscoelastic region. The loss factor tan δ was calculated as the ratio of G″ and G′. The degree of frequency dependence for G′ was determined by the power-law parameters, and is expressed as follows: G′ = A × ω × B,(5)
where G′ is the storage modulus, ω is the oscillation frequency (Hz), and A is a constant.

### 4.6. In Vivo Oral Phase (OP) and Static In Vitro Gastrointestinal Digestion

OP digestion was carried out according to the method proposed earlier [53]. Each type of gel (~4 g) was chewed 20 times by six healthy people (three men and three women) to simulate the maximum destruction of the sample in the oral cavity according to preliminary testing in which the largest number of chews was 20. Thereafter, the bolus was spat three times into a beaker to reduce sample loss. Immediately after this, 4.0 mL of water was added to the beaker, mixed by shaking, and all the liquid part was separated for analysis. The gel pieces were transferred to a 20 mL jacketed glass vessel (reactor) for further in vitro digestion. 

After OP, gels were sequentially incubated in 4.0 mL of the pre-heated SGF (pH 1.5, 0.08 M HCl, and 0.03 M NaCl), SIF (pH 6.8, 0.05 M KH_2_PO_4_, and 0.05 M NaOH), and SCF (0.07 M KH_2_PO_4_, 0.07 M NaHPO_4_, and pectinase: 1.7 mg/mL) at 37 °C and under continuous shaking (250 rpm) for 2, 4, and 18 h, respectively. Before replacing it with another, the medium was completely separated from the gel by a grid (mesh size 350 μm) and used for analysis. After incubation in simulated colonic fluid, the gel residue was destroyed by heating (95 °C, 10 min) in water (16 mL). The data from SCF and gel residue were summed up.

The contents of SSC, GalA, neutral monosaccharides, and PCs were determined in the fluid after each (OP, SGF, SIF, and SCF) phase of digestion. For this, aliquots (1–2 mL) of incubation medium were taken and centrifuged, and the resulting supernatant was precipitated with a fourfold volume of 96% ethanol. The precipitate was washed twice with 96% ethanol before being dissolved in 3 mL of H_2_O; the resulting solution was used to calculate the molecular weight of soluble polysaccharides and the content of GalA by reacting the sample with 3,5-dimethylphenol in the presence of concentrated H_2_SO_4_ [54]. The alcohol supernatant was used to determine the total amount of sugars using the phenol-sulfur method. The PC content was determined in the incubation medium as described above.

### 4.7. Statistical Analysis

The significance of the differences among the means in the PC content, mechanical parameters, and digestion study was estimated with one-way ANOVA and Fisher’s least significant difference (LSD) post hoc test. Statistical differences with *p*-values lower than 0.05 were considered significant. All calculations were performed using the statistical package Statistica 10.0 (StatSoft, Inc., Tulsa, OK, USA). The data presented were expressed as the means ± SD.

## Figures and Tables

**Figure 1 gels-09-00045-f001:**
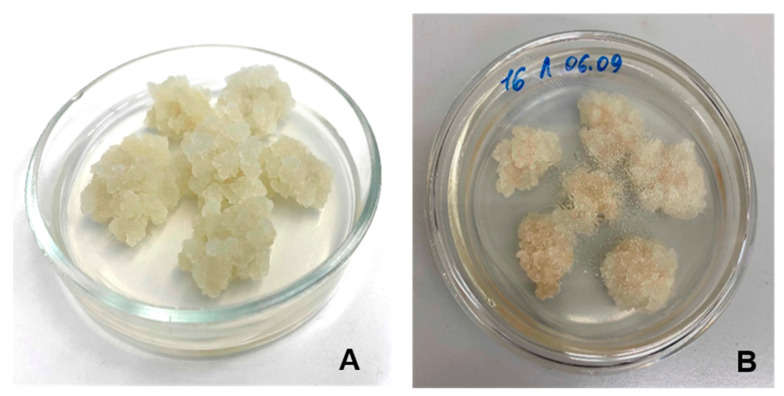
Appearance of the calluses LA14 (**A**) and LA16 (**B**).

**Figure 2 gels-09-00045-f002:**
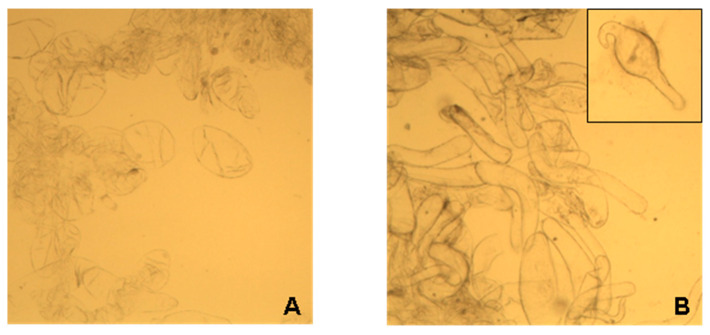
Micrograph of the cells of the callus LA14 (**A**) and LA16 (**B**). Magnification 10×.

**Figure 3 gels-09-00045-f003:**
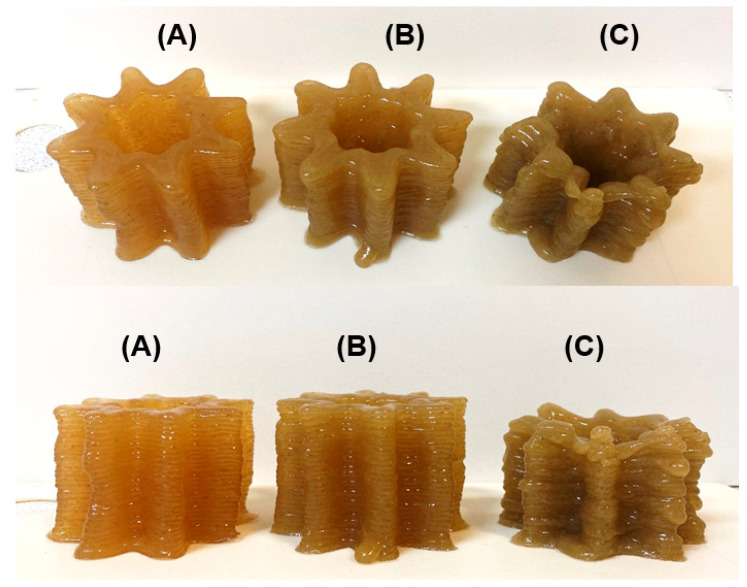
Representative results of the 3D printing experiment using LA14-33 (**A**), LA14-50 (**B**), and LA14-66 (**C**) inks. The top row of photographs is an angled top view; the bottom row is a side view.

**Figure 4 gels-09-00045-f004:**
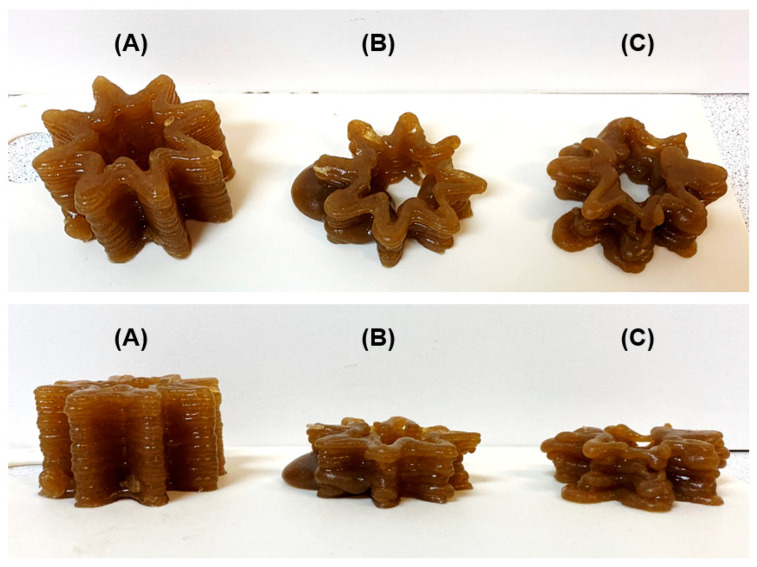
Representative results of the 3D printing experiments using LA16-33 (**A**), LA16-50 (**B**), and LA16-66 (**C**) inks. The top row of photographs is an angled top view; the bottom row is a side view.

**Figure 5 gels-09-00045-f005:**
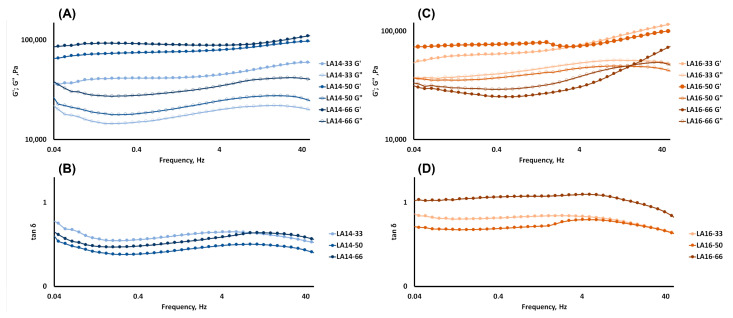
Rheological properties of hydrogels: storage modulus (G′) and loss modulus (G″) test results (**A**,**C**); tan δ test results (**B**,**D**) represented as a function of frequency. Void symbols represent the storage modulus G′, filled symbols represent the viscous modulus G″.

**Figure 6 gels-09-00045-f006:**
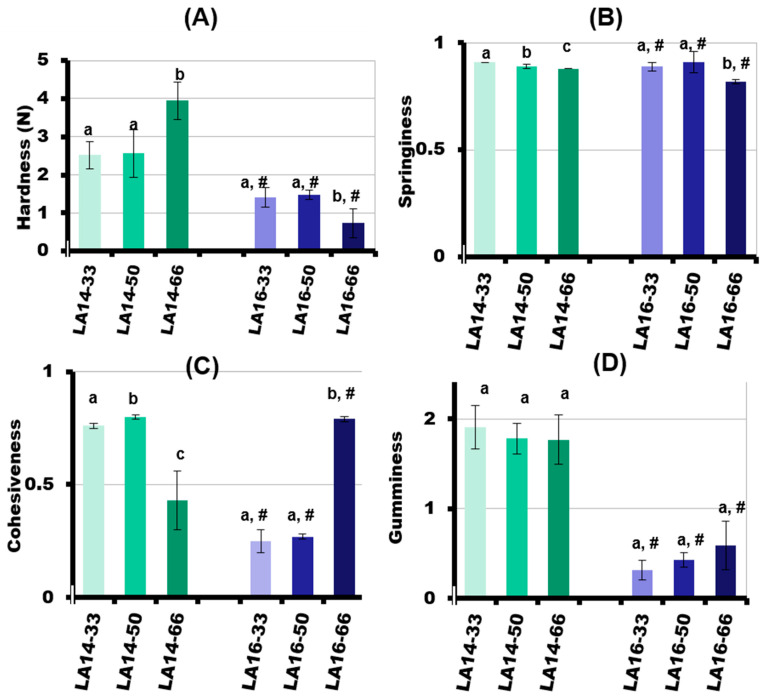
Mechanical properties of the 3D-printed gels. Hardness (**A**), springiness (**B**), cohesiveness (**C**), and gumminess (**D**) are given as the mean ± SD (*n* = 8). Different lowercase letters indicate significant differences (*p* < 0.05) among means for different callus concentrations; #—*p* < 0.05 vs. LA16 callus.

**Figure 7 gels-09-00045-f007:**
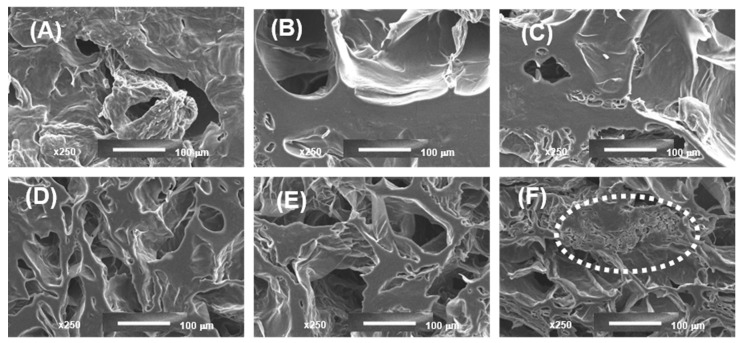
Scanning electron micrographs of the 3D-printed gel containing LA14 (**A**–**C**) and LA16 (**D**–**F**) callus. The content of callus in the ink was 33 (**A**,**D**), 50 (**B**,**E**), and 66 g/100 mL (**C**,**F**). Magnification 250×; scale bar 100 μm. The area of dense folded formation is outlined by a dotted line.

**Figure 8 gels-09-00045-f008:**
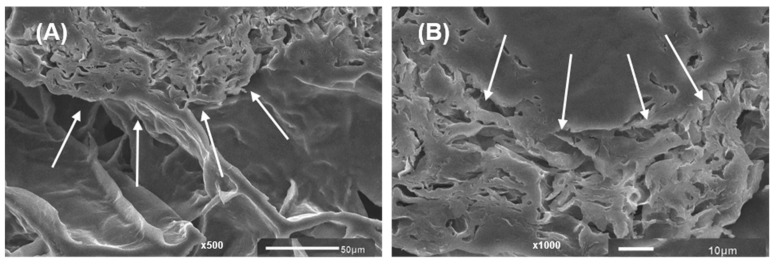
Scanning electron micrographs of the 3D-printed gel containing LA16-66 callus at (**A**) magnification 500×; scale bar 50 μm and (**B**) magnification 1000×; scale bar 10 μm. The arrows indicate the area of dense folded formation.

**Figure 9 gels-09-00045-f009:**
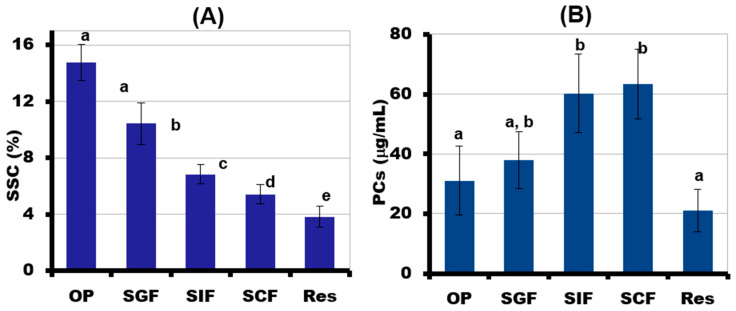
The amount of soluble solids content (SSC) (**A**) and total phenolic compounds (PCs) (**B**) released from LA16-33 gel during successive oral in vivo (OP) and gastrointestinal in vitro phases of digestion. SGF, SIF, and SCF—simulated gastric, intestinal, and colonic fluids, respectively. The data are presented as the mean ± SD. Different lowercase letters indicate significant differences among means for the different phases of digestion (*n* = 6, *p* < 0.05).

**Figure 10 gels-09-00045-f010:**
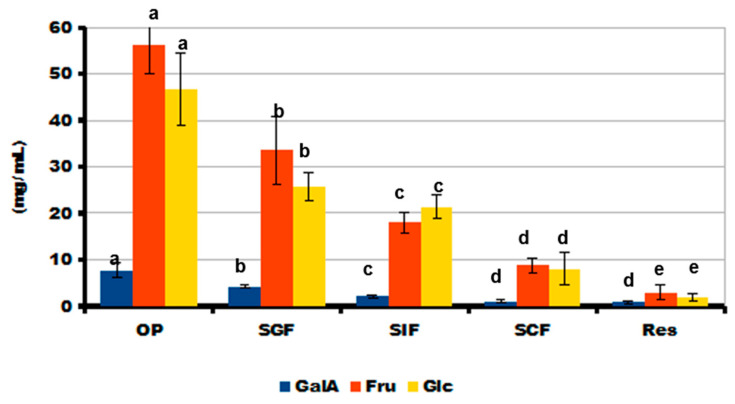
The amount of galacturonic acid (GalA), fructose (Fru), and glucose (Glc) released from LA16-33 gel during successive oral in vivo (OP) and gastrointestinal in vitro phases of digestion. SGF, SIF, and SCF—simulated gastric, intestinal, and colonic fluids, respectively. The data are presented as the mean ± SD. Different lowercase letters indicate significant differences among means for the different phases of digestion of each sugar (*n* = 6, *p* < 0.05).

**Figure 11 gels-09-00045-f011:**
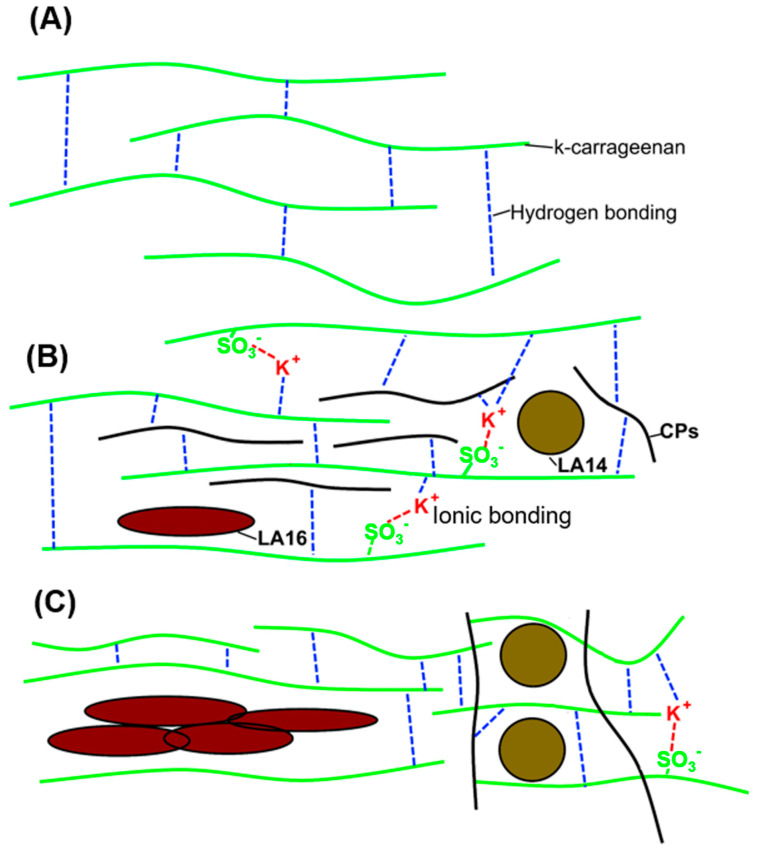
Schematic illustration of a proposed gel network in 3% k-carrageenan (**A**), 3% k-carrageenan enriched with LA callus at 33 g/100 mL (**B**), and 3% k-carrageenan enriched with LA callus at 66 g/100 mL (**C**) inks. CPs–callus polysaccharides. LA14 and LA16–LA14 and LA16 callus cells, respectively. Ionic and hydrogen bonds are shown as red and blue dotted lines, respectively. The polysaccharide chain of k-carrageenan is shown as a solid green line.

**Table 1 gels-09-00045-t001:** The composition of callus of narrow-leaved lupin *L. angustifolius*.

Callus	SSC (%)	Protein ^a^ (%)	EHP ^b^ (%)	HHP ^b^ (%)	PCs ^b^ (%)
LA14	2.68 ± 0.01	0.28 ± 0.01	9.2 ± 0.1	25.0 ± 2.6	1.25 ± 0.02
LA16	4.33 ± 0.11 #	0.30 ± 0.00 #	14.9 ± 0.4 #	30.2 ± 0.4 #	0.53 ± 0.01 #

^a^—per fresh weight; ^b^—per dry matter contents. SSC—soluble solids content; EHP—easily hydrolyzable polysaccharides; HHP—hardly hydrolyzable polysaccharides; PCs—phenolic compounds. The data are presented as the mean ± SD (*n* = 8). #—*p* < 0.05 vs. LA14.

**Table 2 gels-09-00045-t002:** Summary of power law parameters for relationship between storage modulus or viscosity and frequency (0.01 < ω < 50.00 or at 0.1/10.5 Hz) of 3D-printed gel.

Gels	Storage Modulus			Viscosity	
	A (Pa)	B (Slope)	*R* ^2^	η_app_ 0.045, Hz	η_app_ 10.500, Hz
LA14-33	69,196	0.1047	0.926	145,473.3 ± 29,131.1 ^a^	834.9 ± 134.9 ^a^
LA14-50	77,682	0.0355	0.623	300,836.7 ± 18,894.7 ^b^	1702.3 ± 2.9 ^b^
LA14-66	31,914	0.1159	0.603	359,313.3 ± 59,577.9 ^b^	1601.8 ± 324.1 ^c^
LA16-33	43,765	0.0635	0.893	278,873.3 ± 36,029.6 ^a^	1887.0 ± 328.1 ^a^
LA16-50	77,554	0.0474	0.935	162,349.0 ± 30,465.1 ^b^	962.0 ± 189.0 ^b^
LA16-66	92,050	0.0181	0.406	154,626.0 ± 63,583.1 ^b^	855.9 ± 372.0 ^b^

The data are presented as the mean ± SD (*n* = 8). Different letters indicate significant differences (*p* < 0.05) among means for different callus concentrations.

## Data Availability

The data that support the findings of this study are available from the corresponding author upon reasonable request.

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
