# Peer review of "Enrichment of 3D-Printed k-Carrageenan Food Gel with Callus Tissue of Narrow-Leaved Lupin Lupinus angustifolius"

_gels, 2023, doi:10.3390/gels9010045_

Round 1
Reviewer 1 Report
In this MS, the authors, Belova et al. explored the printability of k-carrageenan inks enriched with callus tissue of lupin (L. angustifolius) and to determine the effect of two lupin calluses (LA14 and LA16) on the texture and digestibility of 3D-printed gel. The research was interesting and provide potential good application value of lupin (L. angustifolius). However, there was some questions should be revised. Therefore, I think that the authors should address the following issues:
1. Line 127: Please clearly describe the "LA-66-containing ink" is "LA14-66" or "LA16-66"?
2. As mentioned in Line 108 that "LA14 callus was used only for comparison", but from 3D printability evaluation, LA14 groups had better printing stability than LA16 groups, this meant that LA14 groups were more suitable for 3D printing?
3. Line 158 and Line 159: As mentioned that the G′ of LA16-66 sample was lower than G′′, but as showed in Fig. 5C, the G′ of LA16-66 sample was larger than G′′ at high frequency, how to explain this phenomenon?
4. Fig. 6D: The error bar of LA14-66 was wrong.
5. As mentioned in Line 275 to Line 278, the printability of LA16 groups were lower than LA14 groups. Therefore, why not choose a suitable matrix for LA16 to improve printing precision?
6. From the article could be known, LA16 containing inks had better nutrition, chewing characteristic for elderly and easy digestibility. However, the adding of LA16 would destroy the network of k-carrageenan, and showed lower printing ability. Therefore, how to balance the disadvantage of printing stability and other advantages of LA16?
Reviewer 2 Report
I consider that the manuscript is well structured. adequate methodology and good reference support.. Very interesting and relevant article. The authors show some concern in creating alternative foods based on sustainability when associating k carrageenan food gel with lupine. The conclusions are in line with the studyof lupine calluses LA14 and LA16 incorporated into k-carrageenan-based food ink for 3D printing. The inclusion of lupine calluses in hydrocolloid ink for 3D food printing could facilitate the promotion of gelling products with mechanical properties.
I suggest a revision of the text associated with the legends of figures and tables in general, indicating the true value of p instead of the letters.
Small suggestions for improvement:
Line 257 what is the meaning [9490]
Line 540 confirm EA16? Or LA16?
References 9, 13, … uniform year of publication
Round 2
Reviewer 1 Report
It is accept in this form.